# Analysis of Electric Field Distribution for SOI-FET Sensors with Dielectrophoretic Control

**DOI:** 10.3390/s22072460

**Published:** 2022-03-23

**Authors:** Olga V. Naumova, Elza G. Zaytseva

**Affiliations:** Rzhanov Institute of Semiconductor Physics, Siberian Branch of Russian Academy of Sciences, 630090 Novosibirsk, Russia; zayceva@isp.nsc.ru

**Keywords:** biosensor, field-effect transistor, dielectrophoresis

## Abstract

Silicon-on-insulator (SOI) nanowire or nanoribbon field-effect transistor (FET) biosensors are versatile platforms of electronic detectors for the real-time, label-free, and highly sensitive detection of a wide range of bioparticles. At a low analyte concentration in samples, the target particle diffusion transport to sensor elements is one of the main limitations in their detection. The dielectrophoretic (DEP) manipulation of bioparticles is one of the most successful techniques to overcome this limitation. In this study, TCAD modeling was used to analyze the distribution of the gradient of the electric fields E for the SOI-FET sensors with embedded DEP electrodes to optimize the conditions of the dielectrophoretic delivery of the analyte. Cases with asymmetrical and symmetrical rectangular electrodes with different heights, widths, and distances to the sensor, and with different sensor operation modes were considered. The results showed that the grad *E*^2^ factor, which determines the DEP force and affects the bioparticle movement, strongly depended on the position of the DEP electrodes and the sensor operation point. The sensor operation point allows one to change the bioparticle movement direction and, as a result, change the efficiency of the delivery of the target particles to the sensor.

## 1. Introduction

Electrochemical biosensors play a critical role in the advancement of the point-of-care and lab-on-the-chip systems, for which real-time, simple, highly sensitive, fast analysis, portable sensors are greatly important [1,2]. The biosensors, in which silicon nanochannel field-effect transistors are used, are a versatile platform for such systems. They provide an immediate response, high sensitivity, specificity, and the ability to detect a wide range of bioparticles, including proteins, DNA, RNA, viruses, etc. [3,4,5,6,7]. The typical top-down manufacturing process, the compatibility with silicon-based complementary metal–oxide–semiconductor (CMOS) technology, and, as a result, the reproducibility and easy electronic integration makes Si-based FETs suitable for systems requiring large arrays of multiple functionalized sensors. Moreover, device simulation tools, for example, the TCAD software [8,9] used to design electronic circuits, can be easily adapted to design such biosensors.

The device consists of a Si nanowire or nanoribbon with source–drain contacts at the ends. The sensor, made in a thin-top silicon-on-insulator layer, acts as a dual-gate FET and can operate in two modes—depletion and inversion or accumulation, depending on which carriers are used (minor or major ones). For example, inversion or accumulation is the second mode for n^+^-p-n^+^ or n^+^-n-n^+^ transistors, respectively. The operation mode is provided by the voltage on the SOI substrate that acts as a back gate (BG). The operation point (the voltage on the BG, *V_bg_*) is very important since, in the depletion (also called the subthreshold) mode, the exponential sensor conductivity dependence on the surface potential ensures the maximum sensor response to the detected particles [10]. Any particle adsorbed on the sensor surface changes the surface potential and, as a result, modulates the sensor conductivity due to the field effect (it acts as a second virtual gate). However, at a low analyte concentration in the samples, the bioparticle detection limit is confined by the probability of analyte adsorption on the sensor surface. A simple analytical model, based on reaction–diffusion theory, was developed in report [11] to predict the trade-off between the average response settling time and the minimum detectable concentration for nanobiosensors. It was shown that there exist fundamental limits in the concentration of the biomolecules, which can be detected by any sensor under reasonable settling times in a diffusion-limited regime. The experimental limit of the detection for the nanowire SOI-FET sensors was found to be on the femtomolar level for proteins and DNA [3,4,5,6]. The dielectrophoretic (DEP) delivery of the bioparticles to the sensor elements is one of the most successful methods to overcome this issue [12,13,14,15].

DEP is well known as a technique of manipulating particles resulting from the polarization forces produced by a non-uniform electric field [12]:(1)F→dep= 2πr3εmRe[FCM] ∇|E2|=2πr3εmRe[εp*−εm*εp*+2εm*] ∇|E|2

Here, *r* is the radius of the spherical polarized particle, *ε_m_* is the absolute dielectric permittivity of the medium, Re[*F_CM_*] is the real part of the Clausius–Mossotti factor, the sign of which determines the DEP force direction, |*E*| is the magnitude of the applied field, and $\varepsilon_{p}^{*}$ and $\varepsilon_{m}^{*}$ are the complex dielectric permittivities of the particle and the surrounding medium, respectively.

The DEP force increases with the electric field strength and gradient. The DEP force direction depends only on the relation between $\varepsilon_{p}^{*}$ and $\varepsilon_{m}^{*}$ (see Equation (1)). Under the condition $\varepsilon_{p}^{*}$ > $\varepsilon_{m}^{*}$ (at the positive DEP, p-DEP), the particles are moved towards the maximum electric field. Under the condition $\varepsilon_{p}^{*}$ < $\varepsilon_{m}^{*}$ (at the negative DEP, n-DEP), the particles are repelled away from the highest field-intensity regions. The bioparticle polarizability reflects their uniqueness, which ensures the selective handling of the bioparticles by selecting the appropriate field frequency. Therefore, the DEP forces are widely employed for different platforms with the DEP manipulation of the biological particles.

For a targeted delivery of the bioparticles to the sensor elements, devices with embedded electrodes [13,14] or insulating iDEP structures can be used [15]. In the DEP platforms with embedded electrodes, planar DEP electrodes are typically located near a sensor element. In an electrodeless iDEP platform, an insulating constriction is structured within the sensing-element vicinity, and a voltage is applied to the electrodes on the device periphery [12,15].

For the microfluidic DEP devices, the size (2D or 3D), shape, and position of the electrodes were shown to be very important for the DEP efficiency [16,17,18,19]. Planar 2D microelectrodes are typically fabricated by depositing ∼tens of nanometer-thick metallic layers onto the substrates, whilst microfluidic channels are typically tens of microns high. It was found that the DEP force in a microchannel generated by the planar 2D electrodes decays exponentially along with the channel height, away from the electrode surface [16,17]. The gradient of the electric field square generated by the 3D electrodes is more consistent along with the channel height, i.e., the 3D microelectrodes support a more uniform particle manipulation throughout the channel height direction. The interdigitated electrode geometry is the most common one used in DEP [18]. The electrodes from both parallel and interdigitated electrode arrays are typically rectangular [19]. The maximum values of F_DEP_ are observed near the corners of the electrodes, where the electric field strength and the Δ*E*^2^ factor are maximal. However, despite a large number of experimental and theoretical studies on the *E* between the electrodes [12,18,19], only a few reports are devoted to such a problem as the DEP delivery of an analyte to the FET sensors. Obviously, the nanowire or nanoribbon SOI-FET sensor integrated onto the space between the electrodes creates the non-uniform electric field due to its geometry and operation mode (due to the voltage applied to the back gate).

This study aimed to analyze the distribution of the gradient of the electric field square for the SOI-FET sensors with embedded electrodes to optimize the dielectrophoretic analyte delivery to the sensor element. The commercial 3D-device simulator TCAD, Synopsys Sentaurus, was used for modeling the SOI-FET sensors with the DEP electrodes integrated onto a common sensor chip in the electrolyte solution. Following the reports [8,19,20], the electrolyte region was described as having intrinsic silicon with modified parameters. The cases with asymmetrical and symmetrical rectangular electrodes with different heights and widths, and with different distances between the DEP electrodes and the different sensor operation points were considered.

The results showed that the sensor generated the electrical field gradient, forming the maximum values at the sensor edges (near the concave and convex surfaces). Near the sensor, the grad *E*^2^ factor, which determines the DEP force acting on the movement of the bioparticles, strongly depended on the sensor operation mode and could reach values comparable with those near the DEP electrodes. It was qualitatively shown that the maximum efficiency of control by the bioparticles from the side of the DEP electrodes could be reached at low *V_bg_* voltages in the sensor depletion (subthreshold) mode at the negative DEP and at high *V_bg_* voltages at the positive DEP for the bioparticles.

## 2. Device Architecture and Simulation Method

In this study, the n^+^-p-n^+^- SOI-FET sensors with the DEP electrodes G1 and G2 placed into the electrolyte solution (Figure 1) were simulated using Synopsys Sentaurus Technology Computer-Aided Design (TCAD) tools.

The sensor thickness, width, and length were 30 nm, 500 nm, and 3 µm, respectively. The buried oxide (BOX) thickness was 200 nm. The acceptor concentration in the sensor element was 10^16^ cm^−3^. The donor concentration was 10^20^ cm^−3^ in the contact regions of the drain (D) and source (S) at the ends of the sensor element.

The aluminum DEP electrodes G1 and G2 were located at a distance of *L* = 0.3–2.0 μm from the sensor. The schematic images of the sensors with asymmetrical and symmetric DEP electrodes are shown in Figure 2. Four cases were considered: sensors with asymmetric electrodes with a width of *W_G_* = 0.5 µm (DEP-1) and with symmetric electrodes with a width of *W_G_* = 0.5 µm (DEP-2), *W_G_* = 1.0 µm (DEP-3), and *W_G_* = 2.4 µm (DEP-4). The electrodes had heights *H_G_* equal to 300 nm, 100 nm, 30 nm, or −100 nm (the electrodes embedded into the BOX).

To simulate the current and the electrostatics of the devices, the constant-mobility model, field-dependent mobility (FLDMOB), the Shockley–Read–Hall (SRH) recombination model, and the Fermi model were used. The substrate of the SOI structures was used as the back gate. The source–drain voltage *V_ds_* was 150 mV. The voltages applied to the DEP electrodes G1 and G2 were 4 V and 0 V, respectively. The typical *I_ds_-(V_bg_*) dependence of the sensor is shown in Figure 1b. Low (*V_bg_* = −0.5 V) or high (*V_bg_* = 0.5 V) voltages were applied to the BG to determine the grad *E*^2^ factor in the different SOI-FET sensor modes (depletion or inversion).

The ionic solution was modeled as an intrinsic semiconductor. This was done because the ionic solutions are not included in the set of the standard materials used in TCAD. However, as is well known, the properties of an ion solution are somewhat similar to those of an intrinsic semiconductor [8,20,21]. The intrinsic semiconductor contains mobile thermally generated holes and electrons, while an ionic solution contains mobile cations and anions. The charge distribution in an ionic solution is described by the Poisson–Boltzmann equation. The equation describing the charge distribution in the intrinsic semiconductor agrees very much with the Poisson–Boltzmann equation if (*E_g_*/2−-*qϕ*) is greater than a few thermal energies (*kT*) (here, *E_g_* is the energy bandgap of the semiconductor, q is the elementary charge, and ϕ is the material-electric potential) [20]. For silicon with *E_g_* = 1.12 eV at room temperature, the condition (*E_g_*/2-*qϕ*) >> *kT* is always satisfied in our simulations. Therefore, the ionic solution was defined as intrinsic silicon with the water dielectric constant ε = 78.5. The mobility of the electrons and the holes in the silicon were reduced to reproduce the behavior of the cations and the anions in a solution, respectively. Following the report [21], the maximum mobility values for the holes and the electrons were set as *µ_max,h_* = 4.98 10^−4^ cm^2^ V^−1^ s^−1^ and *µ_max,e_* = 6.88·10^−4^ cm^2^ V^−1^ s^−1^. According to report [8], the effective densities of the states in the conduction band (*N_c_*) and the valence band (*N_v_*) of the semiconductor (used to simulate the electrolyte region) were determined from the equations for the intrinsic density of the charge carriers *n_i_* in a semiconductor:(2)$$n_{i}^{2}=N_{C}N_{V}exp(-E_{g}/kT)$$
and in the solution:(3)$$n_{i}=i_{eff}N_{avo}$$

Here, *N_avo_* is Avogadro’s number (6.022·10^23^ mol^−1^), *k* is the Boltzmann constant, *T* is the temperature, and *i_eff_* is the ionic charge at a molar concentration.

Using expressions (2) and (3), the values of *N_c_* and *N_v_* were estimated to be equal to 1.5∙10^27^ cm^−3^ for 1 mM electrolyte.

## 3. Results

Figure 3 shows the typical electric field distribution for the sensors with asymmetric and symmetric DEP electrodes calculated at a high *V_bg_* value. The E distributions are shown in the **A** cross section (in the plane (*XY*), see Figure 1) at the distance *Z* = 30 nm from the BOX, which equals the sensor height. As expected, the maximum *E* values were observed near the corners of the DEP electrodes. Increased electric field strength values were also observed near the sensor.

Figure 4a shows the E distributions in the B cross section at the same distance from the BOX Z = 30 nm calculated at different BG voltages. The B cross section is the plane (*YZ*) passing through the corners of the DEP electrodes (see Figure 2). Here and below, the plots along the *Y* axis are shown in dimensionless units. The distance between the electrodes and the sensor’s central part was taken as a unity. The *E(Y)* dependences had the maxima near the convex sensor surfaces and near the electrodes. The increase in the *V_bg_* values led to the increase in the *E* values near the sensor.

Figure 4b,c show the distributions of the gradient of the electric field square in the cross sections of B and C, respectively. The C cross section is the plane (*YZ*) passing through the centers of the DEP electrodes (see Figure 2). For comparison, the dependence grad *E*^2^(*Y*) for the device with the DEP-2 electrodes without the sensor element calculated at a high *V_bg_* value is also shown in Figure 4b. Near the convex sensor surfaces, the grad *E*^2^ factor values were: (1) greater than the ones in the central part of the device without the sensor, (2) 2–3 orders of magnitude less than when near the DEP electrodes at a low *V_bg_* value, and (3) comparable with the ones near the DEP electrodes at a high *V_bg_* value, increasing by 1–2 orders of magnitude with an increasing *V_bg_* value. For both the B and C cross sections, the grad *E*^2^ factor distribution near the sensor weakly depended on the DEP electrode width and differed significantly from the case of the DEP electrodes without the sensor.

Figure 5 shows the distributions of the gradient of the electric field square in the B cross section calculated at the different distances *Z* from the BOX for the DEP-2 sensors with different electrode heights *H_G_*. The grad *E*^2^(*Y*) dependences were calculated at the different BG voltages.

The grad *E*^2^ factor weakly depended on the height of the DEP electrodes and sharply decreased with the increase in the distance from the sensor surface.

Figure 6 shows the distributions of the gradient of the electric field square for the DEP-2 sensors with the different DEP electrode heights calculated along the *Z* direction away from the BOX in the B cross section. The grad *E*^2^(*Z*) dependencies were calculated at a high *V_bg_* value. For electrodes with *H_G_* = −100 nm embedded into the BOX, the grad *E*^2^ factor decreased monotonically with an increasing distance from the BOX. The same result was obtained for the case with *H_G_* = 30 nm. For electrodes with *H_G_* = 300 nm, the minimum values of the grad *E*^2^ were observed for *Z~H_G_*/2. The same result was obtained for the case with *H_G_* = 100 nm. Near the sensor, the grad *E*^2^ factor behavior was similar. The minimum of the grad *E*^2^ values was observed for *Z* equal to about half of the sensor height. The maximum values were observed for *Z* = 0 nm and 30 nm, i.e., near the concave and convex surfaces of the sensor. Moreover, the grad *E*^2^ factor was higher for *Z* = 30 nm than the one for *Z* = 0 nm.

Figure 7a shows the grad *E*^2^(*Y*) distributions in the cross sections of B calculated for DEP-2 sensors with different distances (*L*) between the sensor and the DEP electrode. Here, the distance between the electrodes and the sensor was taken as a unity. The grad *E*^2^(*Y*) dependences were calculated at a high *V_bg_* value. Figure 7b shows the grad *E*^2^(*L*) dependences near the sensor calculated at the different *V_bg_* values.

The grad *E*^2^ factor distribution had the typical behavior with the maxima near the DEP electrode and the sensor. The minima were observed in the region between the electrode and sensor (compared with Figure 4 and Figure 5). The grad *E*^2^(*L*) dependences saturated at *L* > 1 mkm. In the submicron range of *L*, a decrease in the *L* values led to an increase in the grad *E*^2^(*Y*) values. Near the sensor surface, the values of the grad *E*^2^ factor tended to the values observed near the DEP electrode and could reach 10^11^–10^12^ V^2^ cm^−3^ for both the high and low *V_bg_* values.

## 4. Discussion

The results showed that the grad *E*^2^ factor distribution had the maxima both near the DEP electrodes and near the sensor, and the minima in the region between the electrode and the sensor (Figure 4, Figure 5 and Figure 7). This behavior of the grad *E*^2^ factor did not depend on the design parameters, position of the DEP electrodes, or the sensor mode, and was drastically different from the case of the DEP electrodes without a sensor. Near the sensor, the grad *E*^2^ values strongly depended on the sensor operation point and practically did not depend on the DEP electrode geometry (the width and the height of the electrodes, Figure 4 and Figure 5). This means that both the DEP electrodes and the sensor generated the electric field gradient. The electrical field generated by the DEP electrode impacted the values of the grad *E*^2^, leading to their increase in the submicron range *L* (Figure 7b).

Near the sensor, at the distance from the BOX equal to the sensor height, the grad *E*^2^ factor was higher than near the sensor base (Figure 6). It was caused by the behavior of the electrostatic potential. The electrostatic potential dropped more quickly to its bulk value when the surfaces were convex, and more gradually when the surfaces were concave. This result agrees with the report [22].

The grad *E*^2^ factor generated by the sensor was increased with the increase in the *V_bg_* values (Figure 4 and Figure 5). At high *V_bg_* values, the grad *E*^2^ factors (the DEP force, respectively) near the sensor and the DEP electrodes were comparable. According to Equation (1), at the positive DEP, the particles are moved towards the maximum of the electric field square gradient. At the negative DEP, the particles are repelled away from the highest field-intensity regions. Since the grad *E*^2^ factor distribution had the maxima both near the DEP electrodes and near the sensor, the DEP forces generated by the sensor and the DEP electrodes are always opposite in direction, as shown schematically in Figure 8. This means that the maximum efficiency of control by the bioparticles from the side of the DEP electrodes can be reached in the sensor depletion (subthreshold) mode at low *V_bg_* values. At the negative DEP, the forces generated by the electrodes deliver the particles to the sensor. At the positive DEP, the space near the sensor is depleted of the bioparticles. At high *V_bg_* values, the forces generated by the sensor could reverse the results. At the n-DEP, the particles are pushed in the region between the electrode and the sensor. At the p-DEP, the particles are localized both on the DEP electrodes and on the sensor surface. Note that these results were experimentally observed at the virus detection by the sensors with DEP control in the study [14]. Note also that in the case of the p-DEP, the maximum efficiency of control by the bioparticles from the side of the DEP electrodes can be reached at a high *V_bg_* voltage. However, part of the target particles will be lost due to their departure towards the DEP electrodes (Figure 8b).

Thus, the sensor operation point could change the bioparticle movement direction under the same conditions (at the positive or negative DEP) and, as a result, change the DEP control efficiency in delivering the target particles to the sensor. For the same values of the Clausius–Mossotti factor (see Equation (1)), the optimal conditions for the dielectrophoretic delivery of the analyte to the sensor are the negative DEP and the subthreshold mode for the SOI-FET sensor with a low *V_bg_* voltage and the positive DEP at high *V_bg_* values.

Note that, according to the current DEP theory, to overcome the dispersive forces associated with the Brownian motion of the proteins, the grad *E*^2^ factor must be equal to~4∙10^15^ V^2^ cm^−3^. However, for different globular proteins, the grad *E*^2^ factor is much smaller (by the 2–4 orders of magnitude) [23]. This means that the grad *E*^2^ factor values of 10^11^–10^12^ V^2^ cm^−3^ observed near the sensor were sufficient to affect the movement of both globular proteins and viruses.

## 5. Conclusions

This study aimed to analyze the distribution of the gradient of the electric field square for the FET sensors to optimize the dielectrophoretic analyte delivery to the sensor element. Cases with asymmetrical and symmetrical rectangular electrodes with different heights, widths, distances between the sensor and the DEP electrodes, and sensor operation points were considered. It was shown that the FET sensor placed between the DEP electrodes in the electrolyte solution drastically changed the grad *E*^2^ factor distribution in the central part of chip compared to the chip with the DEP electrodes without the sensor. The sensor operation point and the position of the DEP electrodes (the distance between the sensor and the DEP electrodes) were key parameters that determined the grad *E*^2^ values near the sensor. Near the sensor surface, the grad *E*^2^ factor could reach the values of 10^11^–10^12^ V^2^ cm^−3^, which are sufficient to impact the movement of both viruses and proteins. The optimal conditions for the dielectrophoretic delivery of analyte to the sensor are the negative DEP and the subthreshold mode for the SOI-FET sensor with a low *V_bg_* voltage and the positive DEP and high *V_bg_* values.

The obtained results make it possible to choose the optimal conditions for the dielectrophoretic delivery of the analyte to the FET sensors (by selecting the sensor operation point and the appropriate DEP field frequency), increase their limit of detection, and reduce the response time. This is important for the development of highly sensitive electronic detectors of the bioparticles and the lab-on-the-chip systems that are based on them.

## Figures and Tables

**Figure 1 sensors-22-02460-f001:**
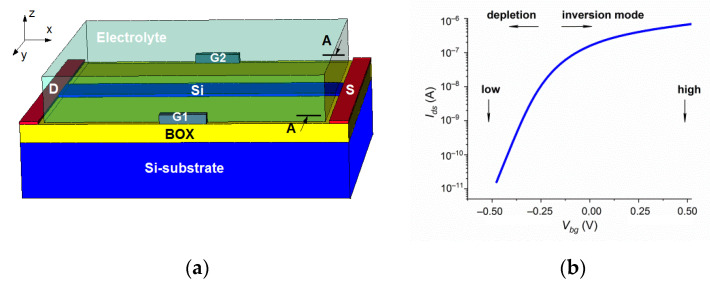
(**a**) Simulated n-channel SOI-FET sensor; (**b**) typical *I_ds_(V_bg_*) dependence of sensors.

**Figure 2 sensors-22-02460-f002:**
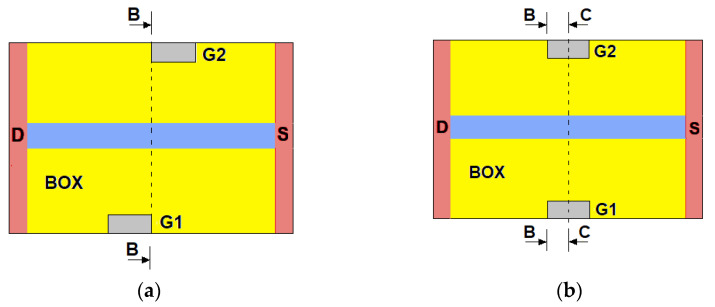
Schematic images of the sensor with **(a**) asymmetric and (**b**) symmetric DEP electrodes.

**Figure 3 sensors-22-02460-f003:**
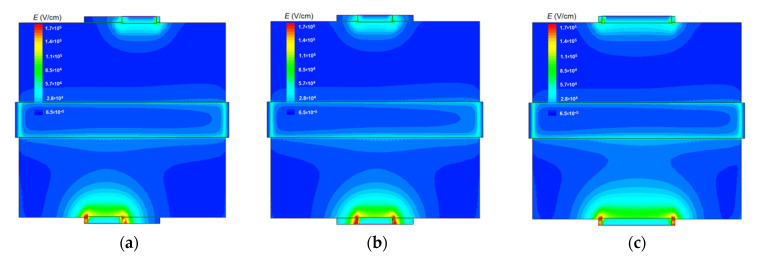
Electric field strength distributions of the A cross section at the distance *Z* = 30 nm from the BOX calculated at *V_bg_* = 0.5 V for (**a**) DEP-1, (**b**) DEP-2, and (**c**) DEP-3 sensors. *H_G_* = 100 nm.

**Figure 4 sensors-22-02460-f004:**
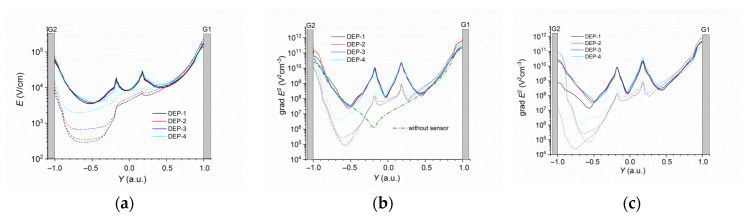
(**a**) Dependencies of *E(Y)*; (**b**,**c**) dependencies of grad *E*^2^*(Y)* calculated at *Z*
**=** 30 nm in the cross sections of (**b**) B and (**c**) C; *V_bg_* = 0.5 V (straight lines) and *V_bg_* = −0.5 V (dashed lines). The dash-dotted line in (**b**) is the case of DEP-2 electrodes without sensor, *V_bg_* = 0.5 V. *H_G_* = 100 nm. *L* = 1.2 mkm.

**Figure 5 sensors-22-02460-f005:**
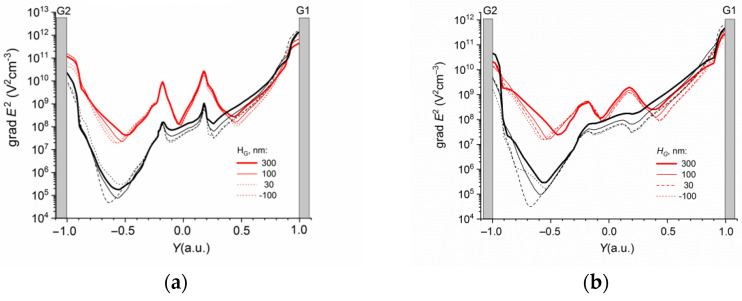
Dependencies of grad *E*^2^
*(Y)* calculated at the distance from the BOX of *Z*, nm: (**a**) 30, (**b**) 100 in the cross sections of B for sensors with different DEP-2 electrode heights *H_G_*. *V_bg_* = −0.5 V (black lines) and *V_bg_* = 0.5 V (red lines). *L* = 1.2 mkm.

**Figure 6 sensors-22-02460-f006:**
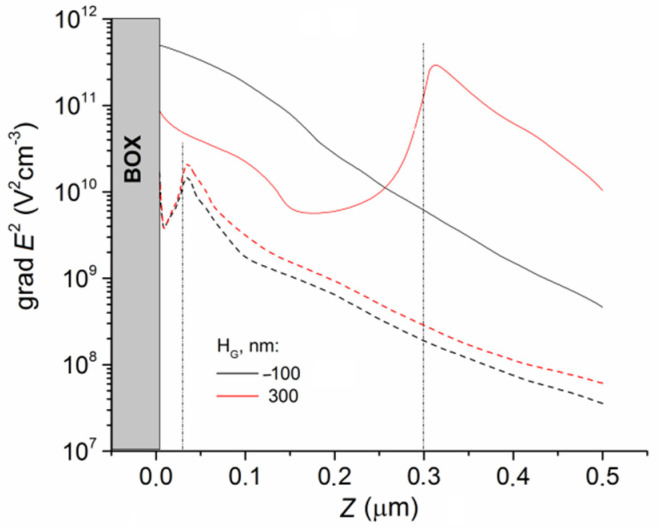
Dependences of grad *E*^2^(*Z*) calculated in the cross section of B near the sensor (dashed lines) and near the DEP-2 electrodes (straight lines) with different heights. *V_bg_* = 0.5 V. *L* = 1.2 mkm.

**Figure 7 sensors-22-02460-f007:**
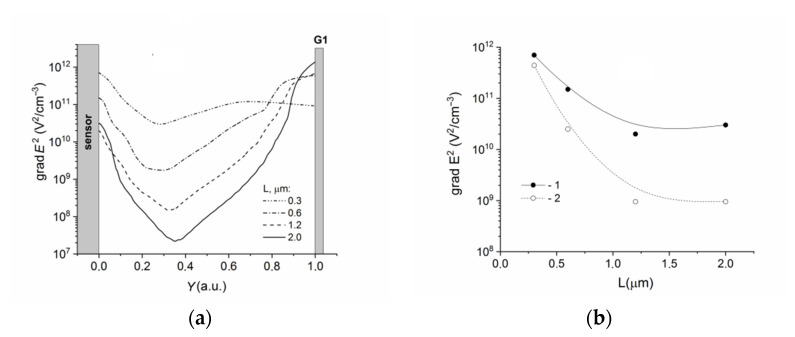
(**a**) Dependencies of grad *E*^2^(*Y*) calculated in the cross sections of B for DEP-2 sensors with different distances *L* to electrode G1. *V_bg_* = 0.5 V. (**b**) Dependencies of grad *E*^2^(*L*) calculated at *Y* = 0 (**a**) and *V_bg_*, V: (1) −0.5 and (2) −0.5. *Z* = 30 nm, *H_G_* = 100 nm.

**Figure 8 sensors-22-02460-f008:**
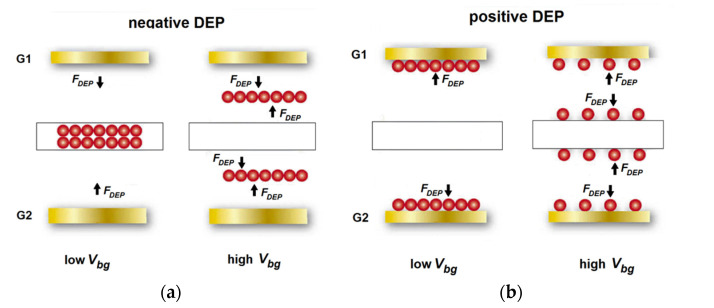
Schematic images of the (**a**) negative DEP and (**b**) positive DEP at different *V_bg_* voltages for the SOI-FET sensor.

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
