# Peer review of "Analysis of Electric Field Distribution for SOI-FET Sensors with Dielectrophoretic Control"

_sensors, 2022, doi:10.3390/s22072460_

Round 1

Reviewer 1 Report

The manuscript aims to provide a "dielectrophoretic control" approach with SOI-FET. Unfortunately, I cannot understand why this concept has a novelty for DEP control.

This study is aimed at analyzing the distribution of the gradient of the electric field square for FET sensors to optimize the dielectrophoretic analyte delivery to the sensor element. 
One of the difficulties in this manuscript is the lack of optimizing trials. The authors proposed the optimized FET sensors for DEP control.
However, the manuscript presented theoretical results of depending on the symmetricity and the height of DEP electrodes. 
I think the authors provided more simulation results depending on each parameter such as electrode width, gate position, etc.
This also requires some rewriting to deliver their message. Unfortunately, I suggest that this manuscript will be rejected.

Author Response

 Dear Reviewer,

 Thanks for Your comments and suggestions.

In this study the simulation results are presented depending on such parameters as symmetry, height of the DEF electrodes, their width (0.5-2.5 μm, Fig. 3) and position (Fig. 7 was added), the sensor operation point.

To deliver the message,

1) In Introduction it was added:

“However, at a low analyte concentration in samples, the bioparticle detection limit is confined by the probability of analyte adsorbtion on the sensor surface…”

“However despite the large number of experimental and theoretical studies for E between electrodes [12, 18, 19], only a few reports are devoted to such problem as the DEP- delivery of an analyte to the FET-sensor”

2) Fig. 7 was added In the Results section

3) in the Discussion section the first paragraph was changed.

4) the Conclusions section with summary was added

Reviewer 2 Report

The manuscript provides an analysis for electric field distribution and dielectrophoretic force for field effect transistor biosensors. The material is presented clearly and the research design is appropriate. Please see my minor comments below:

  1. It will be useful if the plots are provided with non-dimensional distances instead of absolute distances.
  2. The discussion section needs to generalize this approach to utilize it for other similar designs. 

Author Response

Dear Reviewer,

Thanks You very much for Your comments and suggestions

The plots in Figures 4 and  5 are provided with non-dimensional distances instead of absolute distances

To generalize this approach to utilize it for other similar designs  the dependencies of grad E2 (Y) calculated for sensors with different DEP-electrodes position (Fig.7)  were added  and discussed.

“This behavior of the grad E2 factor drastically differs from the case of DEP-electrodes without the sensor …” (in the Discussion section)

Reviewer 3 Report

The authors present dielectrophoretic manipulation of bioparticles as the techniques to solve the diffusion of bioparticles in sensing element. I would suggest the authors include following information before further consideration of the acceptance.

* The limitation of typical SOI based biosensors in the manuscript is not well described. For example, it is difficult to understand the cause of bioparticle diffusion to the sensing element.
* The authors should include perspective, summary, and future direction of the research work in the concluding section. There is no such conclusion in the manuscript.

Author Response

Dear Reviewer,

Thanks You very much for Your comments and suggestions.

We present dielectrophoretic manipulation of bioparticles as a method for solving the problem of diffusion-limited delivery of bioparticles to the sensor (not diffusion).

  1. In introduction it was added:..” at a low analyte concentration in samples, the bioparticle detection limit is confined by the probability of analyte adsorbtion on the sensor surface. A simple analytical model, based on reaction-diffusion theory, was developed in report [11] to predict the trade-off between average response settling time and minimum detectable concentration for nanobiosensors. It was shown that there exist fundamental limits in the concentration of biomolecules which can be detected by any sensor under reasonable settling times in a diffusion limited regime. Experimental limit of detection for nanowire SOI FET-sensors was found to be on the femtomalar level for proteins and DNA [3-6]”.
  2. The Conclusions section was added with summary, perspective and direction research work as experimental.

Reviewer 4 Report

The gradient of E2 between two electrodes is investigated in this article. The study contains no novel findings. There are numerous articles in the literature that developed mathematical models for E between electrodes, including the arrangement of electrodes discussed by the author. Neither the findings nor the methodology are novel. The literature review doesn't discuss key publications in this area. 

Author Response

Dear Reviewer,

Thanks You very much for Your comments and suggestions.

The introduction contains several reviews [ 12, 18, 19 ] with a significant list of publications (more than for hundred). In Introduction it was added: However despite the large number of experimental and theoretical studies for E between electrodes [12, 18, 19], only a few reports are devoted to such problem as the DEP- delivery of an analyte to the FET-sensor”

The results obtained in the work (with using the well-known methodology) are new. The main results (the influence of the operating point on the value of the electric field gradient near the sensor and the behavior of the analyte) are highlighted in the abstract and Discussions section.

The Conclusions section was also added to highlight the results

Reviewer 5 Report

This paper analyzed the distribution of the gradient of the electric field square for SOI-FET sensors. Two different sensors were simulated. Although this work does not actually assemble sensors to verify the simulation results, I still think this paper is of good quality. All the questions I have raised during the review have been well answered in Discussion section. Therefore, I think this paper can be directly accepted for publication after minor revision. The only part that I think the author needs to modify is the absence of the Conlusion section in this paper. I think it is necessary for the author to briefly summarize the results of these simulation calculations after Discussion.

Author Response

Dear Reviewer,

Thank You very much for Your comments and suggestions.

The Conclusions section was added.

Round 2

Reviewer 1 Report

Thank you for your kind response and revision. I think this paper will be published in Sensors.

Reviewer 3 Report

Authors addressed previous comments well.